# Protocol of the Healthy Brain Study: An accessible resource for understanding the human brain and how it dynamically and individually operates in its bio-social context

Healthy Brain Study consortium[1,2,3], Esther Aarts[4], Agnes Akkerman[5], Mareike Altgassen[6], Ronald Bartels[2], Debby Beckers[7], Kirsten Bevelander[2], Erik Bijleveld[7], Esmeralda Blaney Davidson[2], Annemarie Boleij[2], Janita Bralten[2], Toon Cillessen[7], Jurgen Claassen[2], Roshan Cools[8], Ineke Cornelissen[2], Martin Dresler[8], Thijs Eijsvogels[2], Myrthe Faber[8], Guillén Fernández[8]*, Bernd Figner[4,7], Matthias Fritsche[4], Sascha Füllbrunn[5], Surya Gayet[4], Marleen M. H. J. van Gelder[2], Marcel van Gerven[4], Sabine Geurts[7], Corina U. Greven[8], Martine Groefsema[7], Koen Haak[8], Peter Hagoort[3,4], Yvonne Hartman[2], Beatrice van der Heijden[5], Erno Hermans[8], Vivian Heuvelmans[2], Florian Hintz[3], Janet den Hollander[2], Anneloes M. Hulsman[4,7], Sebastian Idesis[9], Martin Jaeger[2], Esther Janse[10], Joost Janzing[2], Roy P. C. Kessels[4,8], Johan C. Karremans[7], Willemien de Kleijn[11], Marieke Klein[2], Floris Klumpers[4,7], Nils Kohn[8], Hubert Korzilius[5], Bas Krahmer[2], Floris de Lange[4], Judith van Leeuwen[8], Huaiyu Liu[7], Maartje Luijten[7], Peggy Manders[2], Katerina Manevska[5], José P. Marques[4], Jon Matthews[2], James M. McQueen[4], Pieter Medendorp[4], René Melis[2], Antje Meyer[3], Joukje Oosterman[4], Lucy Overbeek[2], Marius Peelen[4], Jean Popma[12], Geert Postma[13], Karin Roelofs[4,7], Yvonne G. T. van Rossenberg[5], Gabi Schaap[7], Paul Scheepers[2], Luc Selen[4], Marianne Starren[10], Dorine W. Swinkels[2], Indira Tendolkar[8], Dick Thijssen[2], Hans Timmerman[14], Rayyan Tutunji[8], Anil Tuladhar[8], Harm Veling[7], Maaike Verhagen[7], Jasper Verkroost[2], Jacqueline Vink[7], Vivian Vriezekolk[2], Janna Vrijsen[8], Jana Vyrastekova[5], Selina van der Wal[2], Roel Willems[4,10], Arthur Willemsen[2]

1 Radboud University, Nijmegen, The Netherlands, 2 Radboud University Medical Center, Nijmegen, The Netherlands, 3 Max Planck Institute for Psycholinguistics, Nijmegen, The Netherlands, 4 Donders Institute for Brain, Cognition and Behavior, Radboud University, Nijmegen, The Netherlands, 5 Institute for Management Research, Radboud University, Nijmegen, The Netherlands, 6 Johannes Gutenberg-University Mainz, Germany, 7 Behavioural Science Institute, Radboud University, Nijmegen, The Netherlands, 8 Donders Institute for Brain, Cognition and Behavior, Radboud University Medical Center, Nijmegen, The Netherlands, 9 Center for Brain and Cognition, University Pompeu Fabra, Barcelona, Spain, 10 Centre for Language Studies, Radboud University, Nijmegen, The Netherlands, 11 School of Psychology and Artificial Intelligence, Radboud University, Nijmegen, The Netherlands, 12 Interdisciplinary Hub for Security, Privacy and Data Governance, Radboud University, Nijmegen, The Netherlands, 13 Institute for Molecules and Materials, Radboud University, Nijmegen, The Netherlands, 14 University Medical Center Groningen, Groningen, The Netherlands

* Guillen.fernandez@donders.ru.nl

## Abstract

The endeavor to understand the human brain has seen more progress in the last few decades than in the previous two millennia. Still, our understanding of how the human brain relates to behavior in the real world and how this link is modulated by biological, social, and environmental factors is limited. To address this, we designed the Healthy Brain Study (HBS), an interdisciplinary, longitudinal, cohort study based on multidimensional, dynamic assessments in both the laboratory and the real world. Here, we describe the rationale and

**Funding:** The HBS is funded by the Reinier Post Foundation and Radboud University, Nijmegen, the Netherlands. The funders had and will not have a role in study design, data collection and analysis, decision to publish, or preparation of the manuscript.

**Competing interests:** The authors have declared that no competing interests exist.

design of the currently ongoing HBS. The HBS is examining a population-based sample of 1,000 healthy participants (age 30–39) who are thoroughly studied across an entire year. Data are collected through cognitive, affective, behavioral, and physiological testing, neuro-imaging, bio-sampling, questionnaires, ecological momentary assessment, and real-world assessments using wearable devices. These data will become an accessible resource for the scientific community enabling the next step in understanding the human brain and how it dynamically and individually operates in its bio-social context. An access procedure to the collected data and bio-samples is in place and published on https://www.healthybrainstudy. nl/en/data-and-methods/access.

**Trail registration:** https://www.trialregister.nl/trial/7955.

## Introduction

The human brain is seen as civilization's most precious resource [1], both creating and inter-acting with our increasingly complex environment, it enables us to be conscious and social human beings. Brain functioning also plays a pivotal role in major societal challenges such as health, demographic change, and well-being. Due to developments in different scientific fields, the endeavor to understand the human brain has seen more progress in the last few decades than in the two millennia before. However, we think that current brain research suffers from at least five key limitations and we set up the Healthy Brain Study (HBS) to tackle these five limitations together and, thereby, to facilitate our understanding of how the human brain relates to behavior in the real world and how this link is modulated by biological, social, and environmental factors. In the following paragraphs, we explain the five main design choices of the HBS.

Firstly, a reductionist approach–in which researchers try to understand reality by focusing on a limited number of variables–has been understandably popular as it is vital to obtain detailed mechanistic insights. However, complex dynamical systems, like the human brain, cannot be properly understood by focusing on just one aspect at a time [2–4]. Human brain functioning includes enabling consciousness and cognition, generating emotions, and produc-ing adaptive behavior, and it performs all of these functions while embedded in its biological and social (bio-social) environment [5]. To enable researchers to understand the complexity of human brain functioning in its bio-social context, the HBS provides a broad range of variables within a holistic approach.

Secondly, the brain's operations cannot be fully understood by single assessments obtained at a specific point in time, but require repeated measurements or continuous monitoring. Sin-gle-session assessments may be sufficient to uncover stable traits or processes. However, they do not capture changes in brain functioning that constitute a core feature of our plastic and adaptive brain [6, 7]. Similarly, the body and the social environment are subject to change. For example, seasonality is observed in affect [8, 9], behavior [9, 10], and biological [11–14] and social [9] factors. Most of the studies mentioned were cross-sectional and explicitly stress the need for longitudinal studies that assess within-subject variation. Therefore, in the HBS, par-ticipants perform repeated assessments in three different seasons over one year starting at varying time points within a year. Thereby, we aim to reliably and validly capture changes in human brain operations that may be related not only to seasonality, but also to relevant life events and incidental or dynamic changes in biological factors (e.g., inflammation markers), social factors (e.g., household composition, work relations, friendships, politics, media expo-sure, lockdown), and environmental factors (e.g., daylight hours, exposure to chemicals).

Thirdly, group averages are critical in revealing general principles, but they gloss over differences that make us individual human beings. The human brain is arguably the most individual organ we have and is shaped by our experiences throughout life. Therefore, a large and rich sample is required before single subject inferences can be made about underlying principles of diversity in cognition, affect, and behavior [15, 16]. Given this, the HBS aims to include a broad range of repeated assessments of 1,000 participants.

Fourthly, laboratory assessments enable well-controlled analyses, but they may show low ecological validity in generalizing cognition, affect, and behavior to real-world settings. To understand cognition, affect, and behavior more comprehensively, there is a need for assessments both in the laboratory as well as in the real world [17, 18]. In the HBS, we perform a real-world assessment of physical activity, stress, and sleep with validated wearable devices. Furthermore, we apply ecological momentary assessments using a smartphone application. Taken together, these assessments enable us to understand cognition, affect, and behavior in the context where they naturally occur.

Finally, a healthy volunteer selection bias is a frequent problem in both cohort studies and neuroscience studies. For example, UK Biobank participants were more likely to be female, have a healthy lifestyle, and live in less socioeconomically deprived areas compared to the general population [19]. Also, students, the usual participants in cognitive neuroscience studies, function well, are often relatively healthy and have a high socioeconomic status [20]. Also, most population-based cohorts and large-scale studies include either developing populations [21–23] or advanced aging populations [24–27]. Therefore, the HBS includes a broad population-based sample of individuals who are 30–39 years old that reflects the general population in terms of gender and educational attainment. The age range was chosen to represent adults beyond the age of developmental brain changes and before the onset of brain changes due to advanced aging or neurodegenerative disease. The lower limit of 30 years excludes any neurodevelopment effect as the brain has matured by this point [28]. Also, 30–39 is a socially challenging age range because it is generally characterized by a relatively high number of rather impactful life events (e.g., family planning, career-related changes, buying a house).

In conclusion, the unique feature of the HBS is that it combines the five above-mentioned strengths resulting in in-depth phenotyping of a large range of cognitive, affective, behavioral, and social dimensions with a biological sampling of brain and body-related processes. This enables the extraction of a detailed bio-social fingerprint for the participants in the cohort. Such a detailed fingerprint is currently not available. The availability of HBS will contribute to a better understanding of risks and potentials in behavior in the real world at the individual level. This paper describes the rationale and design of the currently ongoing HBS, which originated from an interdisciplinary, team science [29] based cross-faculty initiative from the Radboud campus in Nijmegen, the Netherlands, including Radboud University, Radboud University Medical Center, and the Max Planck Institute for Psycholinguistics.

## Methods/Design

### Study design and setting

The HBS is a longitudinal cohort study in both laboratory and real-world settings. All laboratory assessments take place at a single-center on Radboud campus, Nijmegen, the Netherlands.

### Participants

The HBS aims to include 1,000 participants (500 men and 500 women) from the Nijmegen region ($\leq$ 15 km) of whom 220 have a low, 340 a middle, and 430 a high level of education. Nijmegen is a medium-sized city in the east of the Netherlands with 176,731 citizens on the 1st

of January 2019 of whom 74% are native Dutch, which is comparable to the overall proportion of native Dutch citizens of the Netherlands (76%) [30]. In contrast, large cities (> 500,000 citizens) in the west of the Netherlands like Amsterdam, Rotterdam, and The Hague have respectively 46%, 48%, 45% native Dutch citizens [30]. Regarding educational attainment, 22% of Nijmegen citizens are primary and secondary educated (low level), 34% are primary, secondary, and vocationally educated (middle level), and 43% of the population have also a university degree (high level). Nijmegen has less citizens with low and middle level of education and more citizens with high level of education compared to the overall proportions of Dutch citizens (28%, 41%, 30% of citizens have respectively low, middle, and high level of education) [30]. In comparison, some large cities in the Netherlands have a higher proportion of citizens with a high level of education (e.g., Amsterdam 48%, Utrecht 52%), while other large cities have a higher proportion of citizens with a low level of education (e.g., Rotterdam 32%, The Hague 31%) [30].

Inclusion criteria are age 30–39 years, living in the Nijmegen region (≤ 15km; because of feasibility), willingness, and ability to follow the study protocol. Exclusion criteria are: not speaking, reading, and/or understanding the Dutch language (minimum B1 level), a prior history of significant psychiatric or neurological illness (self-report), a current disease that affects the brain, a current medication that is therapeutically targeted at the brain (e.g., antidepressants, methylphenidate), pregnancy, contra-indication for MRI (metal or devices in the upper body (cardiac pacemaker, cochlear implant, aneurism clip), previous brain surgery, moderate to severe claustrophobia), contra-indication for the submaximal Åstrand cycle test (current use of beta-blockers, a current disease that hinders physical exercise), contra-indication for the cold pressor test (Raynaud's phenomenon, chronic pain syndrome in shoulder or arm, open wounds on arm or hand, scleroderma, arteriovenous fistula or shunt, presence of (unstable) angina pectoris).

## Recruitment

We aim to acquire full longitudinal datasets of 1,000 participants. We expect a withdrawal rate of 15%, and will therefore recruit 1,150 individuals to participate in the study. We apply different strategies to recruit participants. Firstly, municipalities, general practitioners, and employers of different sectors based in the Nijmegen region send the HBS invitation and research flyer to their citizens, clients, and employees, respectively. Employers are asked to sponsor the study by (partly) exempting their employees from three working days which allows them to participate in three lab visits. Employees remain entirely free to decide whether or not they want to participate. Also, campaigns to increase awareness of the HBS have been launched.

Potential participants fill out contact details in an online form on the website https://www. healthybrainstudy.nl and receive the study brochure. Participants can watch short videos on the website that explain the various tests and assessments or learn about the experiences of an HBS participant. Participants are contacted via phone and invited to a face-to-face information meeting on the Radboud campus. Participants provide written informed consent at this meeting before participation.

## Ethics

The Institutional Review Board of Radboud University Medical Center approved the HBS on the 23rd of May, 2019 (reference number: 2018–4894) in accordance with the latest revision of the Declaration of Helsinki [31]. Incidental findings could occur both while conducting the study (e.g., observed during assessments) and while using the data and biosamples in the future to answer research questions. If a researcher or research assistant notices a potential

finding incidentally, he/she will contact the principal investigator, who approaches an incidental findings committee. At the Radboud Campus, such committees exist for neuroimaging and genetics. For other findings, the principal investigator contacts a medical doctor with relevant expertise. If, according to the committee or medical expert, no clinically relevant finding is identified, the participant remains uninformed. In all other cases, the participant's general practitioner is sent a letter describing the findings. At the same time, the participant receives a request to contact their general practitioner. Participants must consent to this procedure and provide the contact details of their general practitioner, otherwise, they are not allowed to participate.

## Participant panel, feedback of participants, incentives, and citizen science

A participant panel consisting of twelve people (age 30–39, 6 women and 6 men) was set up to aid in the design of the study. The panel advises on communication materials and incentives. For example, the panel gives feedback on the website, study information, posters, and flyers. Moreover, the first 243 participants filled out a questionnaire on their experience of the first lab visit, which provided us with feedback on the study procedures and on keeping participants involved. For example, we developed an online dashboard, because participants indicated that they would prefer more individual feedback on results. Participants receive gadgets after each assessment, and we organize (online) participant events. After completion of the study protocol, participants receive €150 with a maximal addition of €50 for assessment specific incentives.

Besides, a citizen science platform is used to involve participants as well as other citizens in generating research topics and questions that can be investigated with the HBS resource [32]. We 'crowdsource' lists of research topics and/or research questions that participants and citizens think are useful for examining with the HBS resource. At the same time, they also rate the importance of the crowd-generated suggestions by other participants and citizens resulting in an overview that reflects the relevance and prioritization of their overall input.

## Quality management and safety

Research assistants and nurses received extensive training for the assessments undertaken as part of the study protocol. We adapted existing standardized operating procedures (SOPs) if available and developed a new SOP otherwise. An independent study monitor annually performs checks to ensure that the study protocol is followed.

## Data management and data availability

We use Ldot [33], which is a web application that only stores personal and logistical data, for communication with our participants. For data acquisition, we use Castor EDC [34] to provide electronic case report forms and online questionnaires. In addition, a smartphone application for ecological momentary assessments was developed. After participants have performed the real-world assessments, our data managers extract the raw data that is stored locally on the device. Bio-samples are stored at the Radboud Biobank with their sample tracking system, sample processing SOPs, and standardized sample storage conditions being employed [35]. Furthermore, a Polymorphic Encryption and Pseudonymization (PEP) infrastructure was developed for the HBS to protect all data streams and the privacy of participants [36, 37] (Fig 1). Ldot, Castor EDC, the smartphone application, and PEP meet the requirements of the European General Data Protection Regulation.

For each participant, the PEP-system generates unique pseudonyms for the different assessments to avoid the coupling of data to an individual participant during the data collection

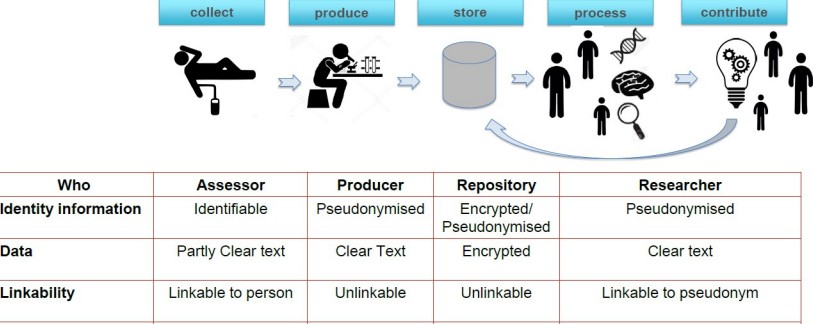

| Who | Assessor | Producer | Repository | Researcher |
|---|---|---|---|---|
| **Identity information** | Identifiable | Pseudonymised | Encrypted/ Pseudonymised | Pseudonymised |
| **Data** | Partly Clear text | Clear Text | Encrypted | Clear text |
| **Linkability** | Linkable to person | Unlinkable | Unlinkable | Linkable to pseudonym |

**Fig 1. The Polymorphic Encryption and Pseudonymization (PEP) infrastructure.**

phase (step 1: collect). A backup of the data is stored locally (step 2: produce) and a copy is encrypted and transferred to the data repository (step 3: store). In the same step, the data are cryptographically pseudonymized. The data can only be decrypted in the processing environment where scientific analyses are performed (step 4: process). The PEP method ensures that different datasets obtained from the repository cannot be linked easily by different research projects because pseudonyms identifying a single participant are personalized at the project level, and data transfer can be minimized based on researchers' requirements. Derived data, produced by researchers, can be stored in the data repository (step 5: contribute) for future use by other researchers using their researcher-specific pseudonyms.

The PEP-system was created to deal with the rigidity of the traditional encryption/decryption process by using polymorphic encryption. PEP ensures that there is no need to a priori fix the encryption key for the data. The PEP system enables different research teams to have access to the entire dataset or only a subset (of participants and variables) of the data source with a specific, personalized decryption key. Due to its additional security, the PEP system is an ideal approach to store, manage, and share sensitive personal data in a research data repository that reduces the risk of a participant's privacy being violated.

## Measures

The following paragraphs describe the measures briefly, while the supplementary information provides detailed descriptions (S1 File). Each assessment starts with pre-visit online questionnaires, followed by a burst week of real-world assessments, followed by a whole day lab visit, which in turn is followed by post-visit online questionnaires and assessments (Fig 2). Only those constructs that may be sensitive to change during one year (states) are repeated during the second and third assessments. The stable (trait) measures are equally distributed over the three assessments. The majority of measures are validated in prior research.

### Pre-visit online questionnaires

Participants fill out questionnaires before the start of the burst week to assess baseline characteristics. The questionnaires cover general demographic questions and questions about lifestyle and well-being (Table 1).

### Burst week with real-world assessments

The burst week consists of a real-world assessment of physical activity, stress, and sleep using validated wearable devices (Table 2) and ecological momentary assessments (EMA) using a

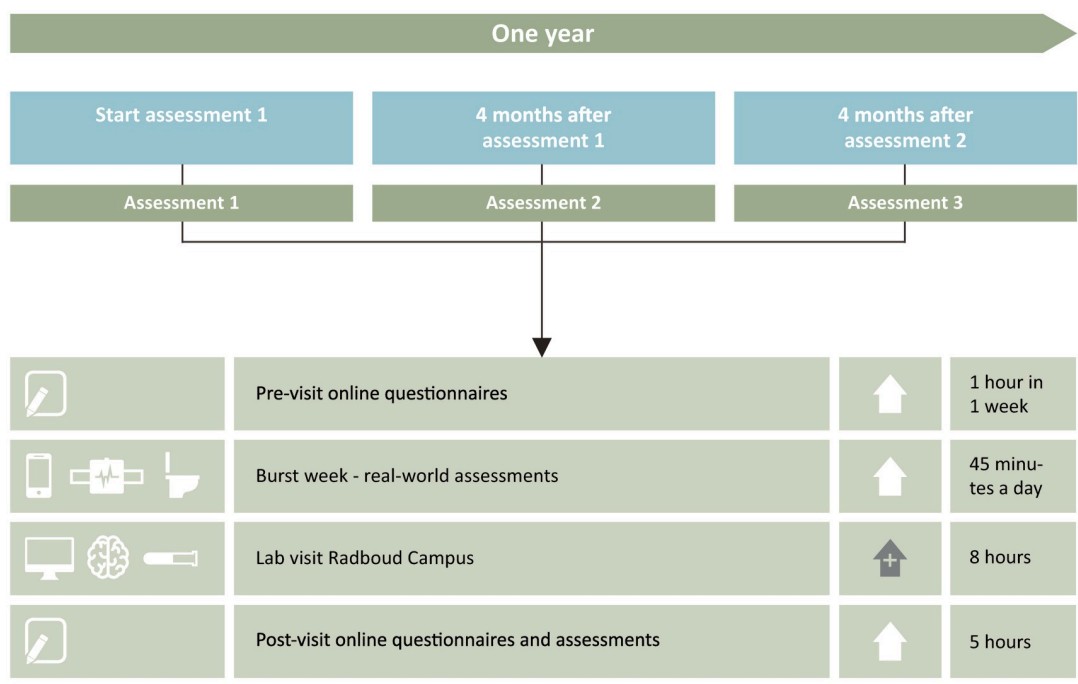

**Fig 2. Design of data collection in the healthy brain study.**

smartphone application. The questionnaire for EMA covers mood, social company, online social interactions, context, control items, retrospection, anticipation, and substance use. In addition, participants perform the home collection of stool, urine, saliva, and diffusive sampling of chemicals using silicone wristbands during the burst week (Table 3).

## Lab visit Radboud campus

Each eight-hour lab visit includes bio-sampling (Table 3), neuroimaging (Table 4), physiological (Table 2), cognitive (Table 5), affective (Table 5), behavioral (Table 5), and sensory assessments (Table 6). To avoid systematic carry-over and fatigue effects, the order of assessments varies between and within participants except for fasting blood sampling and blood pressure at the start of the day.

## Post-visit online questionnaires and assessments

Participants fill out an online questionnaire assessing (mental) health, life events, social/relationships, work, politics, personality, and literacy after each lab visit (Table 7). Also, participants perform several online assessments about decision-making, narrative reading, and solidarity (Table 8). After their third and final lab visit, participants are invited to complete the 'Individual Differences in Language Skills' test battery (Table 9) assessing participants' linguistic knowledge, as well as linguistic processing and general cognitive skills.

**Table 1. Pre-visit online questionnaires.**

| Domain | Name of the questionnaire | What does it measure? | Duration (minutes) | Assessment 1 | Assessment 2 | Assessment 3 | Ref |
|---|---|---|---|---|---|---|---|
| General information | Demographic and socio-economic background | Demographic data, the highest level of education, income, household composition | 10 | x | x | x | [38] |
| | Pregnancy | Number of pregnancies, time to pregnancy, pregnancy outcome, hormones (anticonception), current child wish | 3 | x | | | |
| | Menstrual cycle | Menstrual cycle | 1 | x | x | x | |
| Lifestyle | Smoking history | Past behavior, age of onset | 1 | x | | | |
| | Smoking | Current behavior, frequency, and quantity | 1 | x | x | x | |
| | Fagerstrom Test of Nicotine Dependence (FTND) | Nicotine dependence (for current or ever smokers) | 2 | x | x | x | [39] |
| | Alcohol | Frequency and quantity in the last month, age of onset of alcohol use, binge drinking | 2 | x | x | x | |
| | Alcohol Use Disorder Identification Test (AUDIT) | Heavy alcohol use and associated problems | 3 | x | x | x | [40] |
| | Substance matrix Mate-q | Amount and frequency of substance use | 5 | x | x | x | [41] |
| | Food Frequency Questionnaire (FFQ) | Quantitative food intake | 45 | x | | | [42–45] |
| | Sedentary Behavior Questionnaire (SBQ) | Sedentary behavior in various domains (e.g. home, work, transportation) | 5 | x | x | x | [46] |
| | Pittsburgh Sleep Quality Index (PSQI) | Sleep quality | 5 | x | x | x | [47] |
| | Dream Recall Frequency Scale (DRFS) | Dream recall | 1 | x | x | x | [48] |
| | The Internet Gaming Disorder Scale | Problematic gaming | 2 | x | x | x | [49] |
| | The Social Media Disorder Scale | Problematic social media use | 2 | x | x | x | [50] |
| | Short Media Multitasking Measure (S-MMM) | Use of different media simultaneously | 1 | x | x | x | [51] |
| Well-being | Satisfaction with life scale | Well-being | 2 | x | x | x | [52] |
| | Cantril ladder | Well-being | 1 | x | x | x | [53] |
| | Five Facet Mindfulness Questionnaire–Short Form (FFMQ) | Mindfulness | 10 | x | x | x | [54] |

## COVID-19 questionnaire

From March until July 2020, when the assessment of participants was not allowed due to the COVID-19 measures, the included participants at that point (N = 158) received a monthly questionnaire addressing behavior and worries regarding COVID-19, currently experienced anxiety [94], stress [95], and well-being [53]. Moreover, loneliness [102], sedentary behavior [46], and sleep quality [47] were assessed. We used the same questionnaires as we use in the three repeated assessments (Tables 1 and 7).

## Results—Progress so far

Fig 3 presents the progress and milestones of the Healthy Brain Study. The first participant was included on the 9th of September, 2019.

At the end of June 2021, the HBS included 418 participants. Seventeen-one participants (17%) withdrew from the study so far, mostly because they experienced too much burden

**Table 2. Physiological assessments.**

| Domain | Measure | Location | Assessment 1 | Assessment 2 | Assessment 3 | Ref |
|---|---|---|---|---|---|---|
| **Physical activity** | Fitness | Campus | x | x | x | [55] |
| | Sedentary behavior | Home[1] | x | x | x | [56, 57] |
| **Stress** | Heart rate | Campus | x | x | x | |
| | | Home[1] | x | x | x | [58] |
| | Heart rate variability | Home[1] | x | x | x | [58] |
| | Skin conductance | Home[1] | x | x | x | [58] |
| | Skin temperature | Home[1] | x | x | x | [58] |
| | Startle eye-blink | Campus | x | x | x | [59] |
| | Subjective stress levels | Campus | x | x | x | [60] |
| | | Home[2] | x | x | x | |
| **Sleep** | Sleep duration | Home[1] | x | x | x | |
| | Sleep stages | Home[1] | x | x | x | |
| **Body composition** | Weight | Campus | x | x | x | |
| | Height | Campus | x | x | x | |
| | Waist-hip circumference | Campus | x | x | x | |
| | Body fat | Campus | x | x | x | [61] |
| | Fat weight | Campus | x | x | x | |
| | Total body water | Campus | x | x | x | |
| | Skeletal muscle mass | Campus | x | x | x | |
| | Body fat mass index | Campus | x | x | x | |
| | Fat-free mass index | Campus | x | x | x | |
| **Pain** | Subjective pain levels | Campus | x | x | x | [60, 62] |
| | | Home[1] | x | x | x | |
| | Electrical pain thresholds | Campus | x | x | x | [63, 64] |
| **Cardiovascular** | Blood pressure | Campus | x | x | x | [65] |
| | Carotid artery reactivity | Campus | x | x | x | [66] |

[1] By wearable device,

[2] By ecological momentary assessment (EMA).

(n = 41; 58%), got pregnant (n = 11; 15%), or had been given a diagnosis or medication treatment (n = 8; 11%). Most participants withdrew after the first assessment. To date, participants performed 755 lab visits: 380 participants carried out the first assessment, 237 participants the second assessment, and 138 participants the third assessment completing the entire study protocol.

The COVID-19 pandemic interrupted the data acquisition phase. At that point, the HBS included 158 participants. Due to the lockdown, we canceled all assessments involving physical interaction as of the 16th of March, 2020. The HBS resumed participant assessments on the 15th of July, 2020 in compliance with the directives in force in the Netherlands. As a result, some participants (48%) have more than four months between repeated assessments. Besides, some participants (10%) have a delay between the burst week with real-world assessments and the lab visit at the Radboud campus.

## Discussion

This paper presents the design of the currently ongoing HBS, which will result in a unique and accessible resource for the scientific community and its public and private partners. Data are

**Table 3. Bio-samples and silicone wristband.**

| Bio-sample | Measure | Location | Assessment 1 | Assessment 2 | Assessment 3 | Ref |
|---|---|---|---|---|---|---|
| **Stool** | Gut microbiome | Home | x | x | x | [12, 35, 67] |
| **Urine** (first morning) | Ions, such as calcium, potassium, sodium, magnesium | Home | x | x | x | [35, 68] |
| **Saliva** | Cortisol levels (short term; two baseline samples) | Home | x | x | x | [69] |
| | Cortisol levels (short term; before, immediately after, and 20 minutes after acute challenge) | Campus | x | x | x | |
| **Blood—EDTA plasma** | DNA | Campus | 6 ml* | | | [35] |
| **Blood—PAX gene** | RNA | Campus | 3x 2,5 ml* | 3x 2,5 ml* | 3x 2,5 ml* | |
| **Blood—EDTA plasma** | Future analyses | Campus | 4x 10 ml* | 4x 10 ml* | 4x 10 ml* | |
| | | | 1x 3 ml* | 1x 3 ml* | 1x 3 ml* | |
| **Blood—serum** | Future analyses (e.g., antibodies, proteomics) | Campus | 10 ml* | 10 ml* | 10 ml* | |
| **Blood—heparin plasma** | Future analyses (e.g., hormones, metabolomics) | Campus | 2x 10 ml* | 2x 10 ml* | 2x 10 ml* | |
| **Hair** | Cortisol levels (long term) | Campus | x | x | x | [70] |
| **Silicone wristband** | Exposure to chemicals in the surrounding environment | Home | x | x | x | [71, 72] |

*The indicated volumes refer to whole blood volumes.

collected through cognitive, affective, behavioral, and physiological testing, neuroimaging, bio-sampling, questionnaires, ecological momentary assessment, and real-world assessment using wearable devices. We believe that the HBS complements other studies–small and large–, which together enable the scientific community to take the next step in understanding the human brain and how it dynamically and individually operates in its bio-social context. Here, we present examples of research opportunities including citizen science, reflect on the HBS design choices and study population, and discuss our data security system which enables future data sharing.

**Table 4. Neuroimaging at the campus.**

| Scan | Description | Duration (minutes) | Assessment 1 | Assessment 2 | Assessment 3 | Ref |
|---|---|---|---|---|---|---|
| **Dummy scanner** | | 10 | x | | | |
| **T1w 3D MPRAGE** | Anatomical scan | 5 | x | x | x | |
| **rfMRI** | Resting-state functional scan followed by resting-state questionnaire | 10 | x | x | x | [73, 74] |
| **mfMRI** | Movie functional scan | 4,5 | x | x | x | |
| **Scout, fieldmap, single-band reference EPIs** | Auxiliary scans | 2 | x | x | x | |
| **Diffusion-weighted imaging scan** | Structural connectivity characterizations and white matter tissue microstructural modelling | 10 | x | | | |
| **High-resolution T1w 3D MP2RAGE anatomical scan** | Quantitative T1 and cortical myelin mapping | 10 | | x | | [75] |
| **High-resolution T2*w scan** | Quantitative T2* and magnetic susceptibility mapping for identification and quantification of iron deposition across the brain | 10 | | | x | [76] |

**Table 5. Overview of cognitive, affective, and behavioral assessments at the campus.**

| Domain | Name of task | Measure | Description | Duration (minutes) | Assessment 1 | Assessment 2 | Assessment 3 | Ref |
|---|---|---|---|---|---|---|---|---|
| **Cognition** | Foraging task | The tendency to explore alternatives vs. to exploit a chosen alternative | Participants are presented with a tree and have to decide whether to harvest it for apples and incur a short harvest delay or move to a new tree and incur a longer travel delay | 30 | x | x | x | [77] |
| **Cognition** | Serial random-dot motion discrimination task | How predictions from the past are weighted with uncertain sensory information in the present | Participants judge the motion direction of moving dots (up vs. down) and receive auditory feedback about the correctness of their response | 25 | x | x | x | [78] |
| **Cognition** | Reward-driven reach-adaptation task | How willing people are to search for more rewarding outcomes in a motor task | Participants make shooting movements toward a target while holding a handle that records pulling and hand rotation movements | 20 | x | x | x | [79] |
| **Cognition** | Paired associate memory task | Associative Memory | Participants memorize the associations between pictures of people and names in a study phase and the memory for these associations is tested in a test phase using a cued-recall-test | 7 | x | x | x | [80] |
| **Cognition** | Tower of London | Executive function (planning) | Participants are presented with a startling array of different colored, same-sized balls and are requested to move the balls one-by-one, with as little moves as possible to a predefined goal array. | 5 | x | x | x | [81] |
| **Affect** | Contextual fear generalization task | Fear generalization | Participants are instructed to attend to the presented stimuli and learn to predict the shock in multiple contexts while assessing eye-blink startle electromyography, subjective report, and avoidance tendencies. | 40 | x | x | x | [82] |
| **Affect** | Emotion regulation task | Emotion regulation | Participants are asked to actively regulate their emotions while either neutral or aversive pictures are presented on the computer screen | 15 | x | x | x | [83] |
| **Affect** | Self-referent encoding Task | Positive and negative memory bias | Participants endorse and memorize positive and negative words | 8 | x | x | x | [84] |
| **Affect** | Stimulus-response compatibility task | Automatic approach or avoidance tendency | Participants are presented with pictures (alcohol vs. soda) and are instructed to approach or avoid a certain condition | 10 | x | x | x | [85] |
| **Behavior** | Columbia card task | Risk preference | A card game that gives participants the repeated choice between risky options and safe options | 22 | x | x | x | [86] |
| **Behavior** | Food auction task | Reliable index of people's preference for hedonic (short-term reward) vs. healthy food (long-term reward) | Participants bid on different food items (e.g., package of M&Ms, apple) | 15 | x | x | x | [87] |

**Table 6. Sensory assessments.**

| Domain | Measure | Duration (minutes) | Assessment 1 | Assessment 2 | Assessment 3 |
|---|---|---|---|---|---|
| **Vision** | Contrast sensitivity | 5 | x | | |
| | Visual acuity | 5 | | x | |
| | Color vision | 5 | | | x |
| **Hearing** | Hearing ability | 1 | x | x | x |

**Table 7. Post-visit online questionnaires.**

| Domain | Name of the questionnaire | What does it measure? | Duration (minutes) | Assessment 1 | Assessment 2 | Assessment 3 | Ref |
|---|---|---|---|---|---|---|---|
| **Exposure** | Exposure | Exposure from environment | 5 | x | x | x | |
| **Health** | Over-the-counter medication | Use of nonprescription medication like pain relievers, cough suppressants, etc. | 1 | x | x | x | [88] |
| | Health complaints | Complaints like tiredness, nausea, back pain, headache, etc. | 5 | x | x | x | [89] |
| **Mental Health** | Adult ADHD Self-Report Scale (ASRS) | Symptom scale for ADHD | 10 | x | | | [90] |
| | Autistic Trait Questionnaire (ATQ) | Autistic traits | 5 | x | | | [91] |
| | Self-Report Inventory of Depressive Symptomatology (IDS-SR) | Presence and severity of depressive symptoms | 5 | x | x | x | [92] |
| | Anxiety Sensitivity Index (ASI) | Anxiety (trait) | 5 | | | x | [93] |
| | State and Trait Anxiety Inventory (STAI-S) | Anxiety (state) | 5 | x | x | x | [94] |
| | Perceived Stress Scale (PSS) | Stress | 5 | x | x | x | [95] |
| | Utrecht Burnout Scale (UBOS) | Burnout | 3 | x | x | x | [96] |
| | Reactive Proactive Aggression Questionnaire (RPQ) | Aggression | 5 | x | x | x | [97] |
| | Daily hassles | Daily hassles | 5 | x | x | x | [98] |
| | Cognitive emotion regulation questionnaire (CERQ) | Cognitive regulation of emotion | 5 | x | x | x | [99] |
| **Life events** | Childhood Trauma Questionnaire (CTQ) | Adverse childhood experiences | 5 | | x | | [100] |
| | Life events | Threatening life experiences | 10 | x | x | x | [101] |
| **Social/ Relationship** | UCLA loneliness scale | Loneliness | 5 | x | x | x | [102] |
| | Need to belong scale | Belongingness | 3 | x | x | x | [103] |
| | Multidimensional scale of Perceived Social Support (PSS) | Perceived social support | 5 | x | x | x | [104] |
| **Work** | Exposure to work | Working hours, working schedules, type of employment | 4 | x | x | x | |
| | Survey Work-home Interaction–NijmeGen (SWING) | Work-life balance | 4 | x | x | x | [105] |
| | Workplace commitment | | 5 | x | x | x | [106] |
| | Employability | | 5 | x | x | x | [107, 108] |
| | Questionnaire on the Experience and Evaluation of Work (QEEW) | Job characteristics | 7 | x* | | | [109] |
| **Politics** | Populism index | Attitude toward populism | 2 | x | x | x | |
| | Political efficacy | Attitude towards national government and politics | 2 | x | x | x | [110] |
| | Political participation | Political activities | 1 | x | x | x | |
| | EU membership | Attitude towards EU membership | 1 | x | x | x | [111] |

(*Continued*)

**Table 7.** (Continued)

| Domain | Name of the questionnaire | What does it measure? | Duration (minutes) | Assessment 1 | Assessment 2 | Assessment 3 | Ref |
|---|---|---|---|---|---|---|---|
| Personality | BIG-5 NEO-FFI-3 | Openness to experience, conscientiousness, neuroticism, extraversion, and agreeableness | 10 | x | | | [112] |
| | Sensory Processing Sensitivity (SPS) | High sensitivity | 5 | | | x | [113] |
| | Barratt Impulsiveness Scale (BIS-11) | Impulsiveness | 10 | | x | | [114] |
| | Self-control | | 10 | x | | | [115] |
| | New general self-efficacy scale | Self-efficacy | 5 | | x | | [116] |
| | Dispositional greed | Greediness | 3 | | | x | [117] |
| | Dark triad | Narcissism, Machiavellianism, psychopathy | 5 | x | | | [118] |
| | Social investment attitudes | Attitudes toward corporate social responsibility | 5 | | x | | [119] |
| Literacy | Numeracy test | Mathematical abilities | 12 | | x | | [120] |
| | Financial literacy | Financial attitudes, skills | 20 | | | x | [121] |
| | Graph literacy | Ability to understand the meaning of graphs | 10 | x | | | [122] |
| | Cultural intelligence | Ability to relate and work effectively across cultures | 2 | | | x | [123] |

*Participants fill out their job characteristics at the first assessment. In the second and third assessments, they fill out their job characteristics only in case of a new job.

## Examples of research opportunities

The HBS resource will be used to address expert and citizen-driven research questions that usually pertain to complex interactions between multiple factors. The first example of an expert-driven research question pertains to the association between income and positive affect. It was found among US residents that higher income was associated with more happiness and enjoyment, and less sadness and worry, but only up to a point ($75.000 per year), above that,

**Table 8. Post-visit online assessments.**

| Domain | Online task | What does it measure? | Duration (minutes) | Assessment 1 | Assessment 2 | Assessment 3 | Ref |
|---|---|---|---|---|---|---|---|
| Decision-making | Higher-order risk preferences | Risk attitudes, prudence, and temperance in financial decision-making | 15 | x | x | x | [124] |
| | Equality equivalence test | Social preferences | 10 | x | x | x | [125] |
| | Ambiguity | Ambiguous risk attitudes | 10 | x | x | x | [126] |
| | Trust game | Trust and trustworthiness | 10 | x | x | x | [127] |
| | Public good game | Altruism, conditional reciprocity | 15 | x | x | x | [128] |
| | Time preferences | Temporal discounting | 8 | x | x | x | [129] |
| Language | Narrative reading | Comprehension of and immersion into a narrative | 15 | x | x | x | [130] |
| Solidarity | Vignettes | Culpability, in/out group | 15 | x | x | x | [131, 132] |

**Table 9. Individual differences in language skills test battery.**

| Domain | Online task | What does it measure? | Duration (minutes) | Ref |
|---|---|---|---|---|
| **Cognition** | Auditory simple and choice reaction time task | Processing speed | 7 | [133] |
| | Letter comparison | Processing speed | 5 | [134, 135] |
| | Visual simple and choice reaction time task | Processing speed | 7 | [133, 136] |
| | Digit span (forward & backward) | Auditory working memory | 7 | [137] |
| | Corsi block tapping (forward & backward) | Visual working memory | 7 | [138, 139] |
| | Raven's advanced progressive matrices | Non-verbal intelligence | 25 | [140] |
| **Linguistic knowledge** | Stairs4Words (2 Runs) | Linguistic experience: Vocabulary | 7 | |
| | Peabody Picture Vocabulary Test | Linguistic experience: Vocabulary | 10 | [141, 142] |
| | Idiom recognition test | Linguistic experience: Knowledge of idiomatic expressions | 3 | |
| | Spelling test | Linguistic experience: Spelling | 5 | |
| | Author recognition test | Linguistic experience: Print exposure | 5 | [143] |
| | Prescriptive grammar | Linguistic experience: Prescriptive grammar knowledge | 10 | [144] |
| **Linguistic processing** | Picture naming | Word production | 7 | [133] |
| | Rapid automatized naming | Word production | 7 | |
| | Verbal fluency | Word production | 5 | [145] |
| | Antonym production | Word production | 5 | [146] |
| | Maximal speech rate | Word production | 3 | |
| | Phrase generation | Sentence production | 10 | |
| | Sentence generation (active/passive sentence formulation) | Sentence production | 12 | |
| | Sentence generation (event apprehension) | Sentence production | 10 | |
| | Spontaneous speech | Sentence production | 4 | [147] |
| | Non-Word monitoring in non-word lists in noise | Word comprehension | 10 | |
| | Rhyme judgment | Word comprehension | 5 | |
| | Lexical decision | Word comprehension | 7 | [133] |
| | Semantic categorization | Word comprehension | 5 | |
| | Word monitoring in sentences in noise | Sentence comprehension | 10 | |
| | Grammatical gender cues | Sentence comprehension | 10 | [148] |
| | Verb-specific selective restrictions | Sentence comprehension | 7 | [149, 150] |
| | Self-paced reading | Sentence comprehension | 5 | |

there was no relationship between income and emotional well-being [151]. The HBS resource can help explain the interplay between affect, social and biological data, and income. A second example of a complex interaction is that sedentary behavior is associated with poor health and higher mortality [152, 153]. Merely standing up from time to time, e.g., to walk around a bit protects against part of this health risk [154]. Existing research on this topic has mainly focused on the consequences of prolonged sitting and has overlooked the key question of why people choose to stand up (when they sit) or sit down (when they stand), in the first place. In other words, what psychological processes (e.g., related to effort, reward, affect, and fatigue) are associated with healthy and unhealthy sedentary behavior? Answering this question will pave the

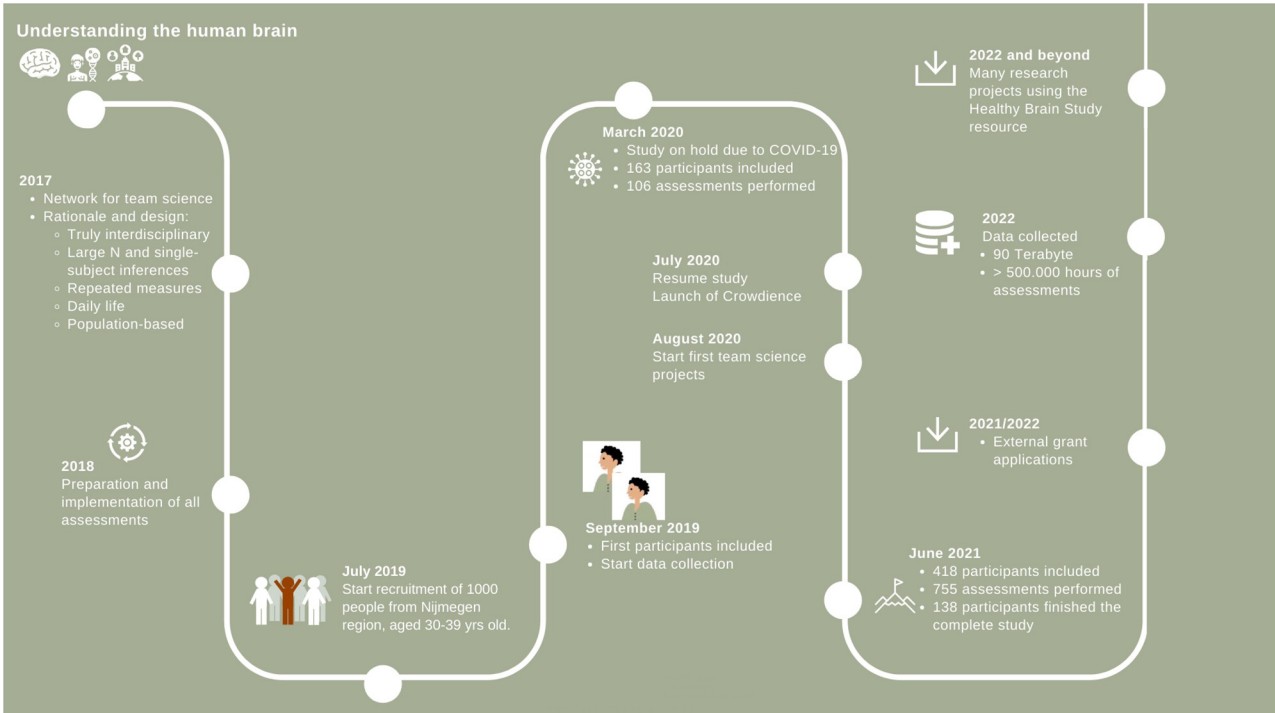

**Fig 3. Progress and milestones of the healthy brain study.**

way for the development of novel, targeted interventions that will improve (occupational) health [155].

The HBS resource will also be used for citizen science. Different forms of citizen science exist. Projects can be led by experts, community-led, or co-created with different aims and levels of participation [156]. HBS participants and other citizens generate research questions to be answered with the HBS resource. In traditional designs, scientists test hypotheses that are often based on previous findings within their research domain or their intuitions. However, people living in or with specific conditions (i.e., being in their thirties and going through a key life event) may have additional insight on top of existing expert-knowledge. These insights are uncovered by a citizen science platform. The essence of the platform is to leverage collective intelligence from a large group of participants versus a smaller number of experts. This can reveal topics and research questions that have a significant influence on people's behavior in the real world and their health status, which experts may have left untouched [157–160]. By giving citizens a voice in scientific research, it can contribute considerably to the valorization of research results.

### Reflection on design choices and study population

Comprehension of complex interactions as illustrated above requires an interdisciplinary, team science approach [29]. The HBS design is the result of an orchestrated cross-campus process over 22 months in which 250 scientists from all classical faculties were involved and were challenged to look past the horizons of their disciplines in a few plenary meetings and several smaller working groups, all providing input to a multidisciplinary scientific board that made the final design decisions. Here, we reflect on our design choices and the study population selected.

First of all, to capture the complexity of the human brain and its environment, a large set of measures was provided. We sought a balance between comprehensiveness, local expertise, costs, and burden for the participants. This resulted in an extensive number, variety, duration of mostly validated assessments, albeit not perfectly comprehensive. For example, the neuro-imaging protocol is largely aligned with the Human Connectome Project [161] and UK Bio-bank [162] brain imaging, but includes a movie fMRI scan that is not included in the Human Connectome Project and UK Biobank, while the latter include scans that are not included in the HBS. Furthermore, instead of continuous monitoring over one year with validated wearable devices, the HBS covers three times a burst week of real-world assessments. Also, the collection of GPS data, financial transactions, and social media interactions were not allowed due to legal restrictions and privacy concerns.

Secondly, the HBS includes three repeated assessments for about one year. These repetitions aim to capture changes in human brain operations that may be related to relevant life events, seasonality, and/or incidental or dynamic changes in the biological and social environment. Regarding seasonality, the HBS participants start at varying time points within a year, so, although we sample only three instead of four times over one year, across participants we sample seasonal transitions in a fine-grained manner.

Thirdly, the HBS aims to include 1,000 participants. Due to differences in measurement-specific signal-to-noise properties, it is not possible to provide a straightforward power and sample size calculation because the data enables analyses of various cognitive, affective, and behavioral interactions in their bio-social context. To decide on the number of participants, we sought a balance between sensitivity and feasibility. The chosen number of participants is high compared to traditional neuroscientific experiments revealing general principles but is low compared to disease risk-oriented cohorts (which is not the aim of the HBS) like the UK Biobank [26], the Rotterdam Study [24], or the Rhineland Study [25]. However, the number is comparable to other studies designed to capture inter-individual differences like the Human Connectome Project, which included 1,200 young healthy adults [161], or the Personalized Parkinson Project, which included 650 patients [163]. We consider the number large given the comprehensive range of repeated measures both in the laboratory as in the real world.

Fourthly, we believe that assessing real-world events with wearable devices is more objective than scales and questionnaires. When we designed the study in 2017, hardly any longitudinal study included wearable devices. As the field of wearable technology has developed rapidly, in the meantime, several longitudinal studies have added wearable devices to their data collection methods. For example, subsets of UK Biobank participants and Rotterdam Study participants wore an accelerometer [164, 165]. We would like to stress that including real-world assessments is one of the five strengths of the HBS, so it is not unique by itself. In particular, the HBS includes repeated assessments with wearable devices in three different seasons over one year starting at varying time points within a year. When we designed the study, to our knowledge, this was unique for HBS. In the meantime, a subset of UK Biobank participants is performing seasonal repeats with a wearable device [166]. Furthermore, the HBS combines physiological recordings with wearable devices with ecological momentary assessments using a smartphone application. We consider the additional collection of momentary assessments of mood and behavior and context information innovative.

Fifthly, we developed a recruitment strategy targeted at a sample that represents the 30-39-year-old population of Nijmegen and its surroundings in terms of gender and educational attainment. However, a reasonable level of reading, speaking, and understanding Dutch (B1 level) is required to be able to complete the study protocol, e.g., to fill out questionnaires. This implies that the HBS participants do not fully represent the Nijmegen population at large, because in this example the illiterate, people with low literacy, or non-Dutch speaking

individuals are excluded. However, the aim of including 220 participants with a low, 340 with a middle, and 430 with a high level of education enables the study of interacting social factors.

## Digital security system and data sharing

The HBS resource will be accessible to the scientific community at large. The resource contains sensitive personal data that needs to be protected from unauthorized access and unintentional disclosure. The sharing of (big) data within the scientific community is necessary for progress and maximizes scientific benefits derived from valuable and costly data. The HBS data is protected by a digital security system, a Polymorphic Encryption and Pseudonymization (PEP) infrastructure [36], which allows the sharing of data with researchers worldwide while safeguarding participants' privacy in line with the European General Data Protection Regulation. The digital security system is based on a multi-point, privacy-by-design strategy: (a) participants provide informed consent, also for the important element of data sharing; (b) signed contractual agreements with researchers are in place to ensure that no attempts towards de-pseudonymization, linking or commercialization of the raw data will be attempted; (c) governance policies limit access to the data to qualified researchers only; (d) an innovative pseudonymization and encryption process is applied [37].

An access procedure is in place and published on https://www.healthybrainstudy.nl/en/data-and-methods/access. We stratify researchers into three tiers with different rights. Tier I consists of researchers from the Radboud campus that contributed to study design or data acquisition. Tier II consists of all other researchers from the Radboud campus. Tier III consists of publicly financed researchers from other academic institutions. Companies can apply in all tiers, but they cannot apply independently. Application for data starts with the submission of a data request for a project that has been preregistered, e.g., in the Open Science Framework. Then, the HBS scientific board reviews the application. After approval, the researcher signs a data/material transfer agreement. Next, the researcher receives data and/or samples. The Radboud Biobank provides the samples [35]. All processed data and samples with relevant documentation (including scripts and data and/or samples processing protocols) must be integrated back into the HBS resource so that it can be used by others. Finally, the researcher publishes the results by acknowledging the HBS consortium.

## Conclusion

The HBS has been designed using a team science approach to integrate scientific disciplines and is characterized by a broad range of repeated assessments, a large number of participants, both laboratory and real-world assessments, and a population-based sample. Moreover, data is managed to allow data sharing with scientists worldwide while maintaining participants' privacy. With the HBS resource, the scientific community can take the next step in understanding the human brain and how it dynamically and individually operates in its bio-social context.

## Supporting information

**S1 File. Detailed descriptions of measures included in healthy brain study.**
(DOCX)

## Acknowledgments

Preprint DOI: 10.31219/osf.io/fqmdu.

## Author Contributions

**Conceptualization:** Esther Aarts, Agnes Akkerman, Mareike Altgassen, Ronald Bartels, Debby Beckers, Kirsten Bevelander, Erik Bijleveld, Esmeralda Blaney Davidson, Annemarie Boleij, Janita Bralten, Toon Cillessen, Jurgen Claassen, Roshan Cools, Martin Dresler, Thijs Eijsvogels, Myrthe Faber, Guillén Fernández, Bernd Figner, Matthias Fritsche, Sascha Füllbrunn, Surya Gayet, Marleen M. H. J. van Gelder, Marcel van Gerven, Sabine Geurts, Corina U. Greven, Martine Groefsema, Koen Haak, Peter Hagoort, Yvonne Hartman, Beatrice van der Heijden, Erno Hermans, Florian Hintz, Anneloes M. Hulsman, Martin Jaeger, Esther Janse, Joost Janzing, Roy P. C. Kessels, Johan C. Karremans, Willemien de Kleijn, Marieke Klein, Floris Klumpers, Nils Kohn, Hubert Korzilius, Floris de Lange, Judith van Leeuwen, Huaiyu Liu, Maartje Luijten, Peggy Manders, Katerina Manevska, José P. Marques, James M. McQueen, Pieter Medendorp, René Melis, Antje Meyer, Joukje Oosterman, Lucy Overbeek, Marius Peelen, Geert Postma, Karin Roelofs, Yvonne G. T. van Rossenberg, Gabi Schaap, Paul Scheepers, Luc Selen, Marianne Starren, Dorine W. Swinkels, Indira Tendolkar, Dick Thijssen, Hans Timmerman, Rayyan Tutunji, Anil Tuladhar, Harm Veling, Maaike Verhagen, Jacqueline Vink, Janna Vrijsen, Jana Vyrastekova, Selina van der Wal, Roel Willems, Arthur Willemsen.

**Data curation:** Willemien de Kleijn, Jasper Verkroost.

**Funding acquisition:** Guillén Fernández, Peter Hagoort, Arthur Willemsen.

**Investigation:** Ineke Cornelissen, Vivian Heuvelmans, Jon Matthews.

**Methodology:** Esther Aarts, Agnes Akkerman, Mareike Altgassen, Debby Beckers, Kirsten Bevelander, Erik Bijleveld, Esmeralda Blaney Davidson, Annemarie Boleij, Janita Bralten, Toon Cillessen, Jurgen Claassen, Roshan Cools, Ineke Cornelissen, Martin Dresler, Thijs Eijsvogels, Myrthe Faber, Bernd Figner, Matthias Fritsche, Sascha Füllbrunn, Surya Gayet, Marleen M. H. J. van Gelder, Marcel van Gerven, Sabine Geurts, Corina U. Greven, Martine Groefsema, Koen Haak, Yvonne Hartman, Beatrice van der Heijden, Erno Hermans, Florian Hintz, Anneloes M. Hulsman, Sebastian Idesis, Martin Jaeger, Esther Janse, Joost Janzing, Roy P. C. Kessels, Johan C. Karremans, Willemien de Kleijn, Marieke Klein, Floris Klumpers, Nils Kohn, Hubert Korzilius, Bas Krahmer, Floris de Lange, Judith van Leeuwen, Huaiyu Liu, Maartje Luijten, Peggy Manders, Katerina Manevska, José P. Marques, James M. McQueen, Pieter Medendorp, René Melis, Antje Meyer, Joukje Oosterman, Lucy Overbeek, Marius Peelen, Geert Postma, Karin Roelofs, Yvonne G. T. van Rossenberg, Gabi Schaap, Paul Scheepers, Luc Selen, Marianne Starren, Dorine W. Swinkels, Indira Tendolkar, Dick Thijssen, Hans Timmerman, Rayyan Tutunji, Anil Tuladhar, Harm Veling, Maaike Verhagen, Jacqueline Vink, Vivian Vriezekolk, Janna Vrijsen, Jana Vyrastekova, Selina van der Wal, Roel Willems.

**Project administration:** Guillén Fernández, Janet den Hollander, Lucy Overbeek.

**Resources:** Janet den Hollander, Peggy Manders, Jean Popma, Dorine W. Swinkels.

**Supervision:** Ineke Cornelissen, Guillén Fernández, Lucy Overbeek.

**Writing – original draft:** Lucy Overbeek.

**Writing – review & editing:** Esther Aarts, Agnes Akkerman, Mareike Altgassen, Ronald Bartels, Debby Beckers, Kirsten Bevelander, Erik Bijleveld, Esmeralda Blaney Davidson, Annemarie Boleij, Janita Bralten, Toon Cillessen, Jurgen Claassen, Roshan Cools, Ineke Cornelissen, Martin Dresler, Thijs Eijsvogels, Myrthe Faber, Guillén Fernández, Bernd Figner, Matthias Fritsche, Sascha Füllbrunn, Surya Gayet, Marleen M. H. J. van Gelder, Marcel

van Gerven, Sabine Geurts, Corina U. Greven, Martine Groefsema, Koen Haak, Peter Hagoort, Yvonne Hartman, Beatrice van der Heijden, Erno Hermans, Vivian Heuvelmans, Florian Hintz, Janet den Hollander, Anneloes M. Hulsman, Sebastian Idesis, Martin Jaeger, Esther Janse, Joost Janzing, Roy P. C. Kessels, Johan C. Karremans, Willemien de Kleijn, Marieke Klein, Floris Klumpers, Nils Kohn, Hubert Korzilius, Bas Krahmer, Floris de Lange, Judith van Leeuwen, Huaiyu Liu, Maartje Luijten, Peggy Manders, Katerina Manevska, José P. Marques, Jon Matthews, James M. McQueen, Pieter Medendorp, René Melis, Antje Meyer, Joukje Oosterman, Lucy Overbeek, Marius Peelen, Jean Popma, Geert Postma, Karin Roelofs, Yvonne G. T. van Rossenberg, Gabi Schaap, Paul Scheepers, Luc Selen, Marianne Starren, Dorine W. Swinkels, Indira Tendolkar, Dick Thijssen, Hans Timmerman, Rayyan Tutunji, Anil Tuladhar, Harm Veling, Maaike Verhagen, Jasper Verkroost, Jacqueline Vink, Vivian Vriezekolk, Janna Vrijsen, Jana Vyrastekova, Selina van der Wal, Roel Willems, Arthur Willemsen.

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
