## [Decision Letter · Decision Letter 0]

23 Jun 2021

PONE-D-21-15454

Protocol of the Healthy Brain Study: an accessible resource for understanding the human brain and how it dynamically and individually operates in its bio-social context

PLOS ONE

Dear Dr. Fernández,

Thank you for submitting your manuscript to PLOS ONE. After careful consideration, we feel that it has merit but does not fully meet PLOS ONE’s publication criteria as it currently stands. Therefore, we invite you to submit a revised version of the manuscript that addresses the points raised during the review process.

While your manuscript is judged favouribly, there are a few points raised by the reviewer that can help to improve the manuscript.

We look forward to receiving your revised manuscript.

Kind regards,

Amir-Homayoun Javadi, PhD

Academic Editor

PLOS ONE

Journal Requirements:

3. Please amend your authorship list in your manuscript file to include author Lucy Overbeek.

Reviewers' comments:

Reviewer's Responses to Questions

**Comments to the Author**

1. Does the manuscript provide a valid rationale for the proposed study, with clearly identified and justified research questions?

Reviewer #1: Yes

2. Is the protocol technically sound and planned in a manner that will lead to a meaningful outcome and allow testing the stated hypotheses?

Reviewer #1: Yes

3. Is the methodology feasible and described in sufficient detail to allow the work to be replicable?

Reviewer #1: Yes

4. Have the authors described where all data underlying the findings will be made available when the study is complete?

Reviewer #1: Yes

5. Is the manuscript presented in an intelligible fashion and written in standard English?

Reviewer #1: Yes

6. Review Comments to the Author

You may also provide optional suggestions and comments to authors that they might find helpful in planning their study.

Reviewer #1: This Study Protocol describes the ongoing Healthy Brain Study (HBS). I found the description comprehensive and I think the resulting database will be a much used resource. The access procedure is well described and reasonable. I have some comments that the authors may consider in a revision, listed in order of appearance in the ms:

* I agree that a reductionist approach is common, but several longitudinal studies surely include a broad range of variables and a holistic approach; this is not unique for HBS.

* Life events are highlighted. Such events have been explored in several other past studies, and a question is whether the authors in the quite short period of 1 year expect a sufficient number of significant events to substantiate analyses? Here, the timing with the pandemic is very interesting and it is a strength of the study to capture this major event.

* The protocol has an elegant assessment of factors such as physical activity, stress, and sleep that go beyond past studies. However, to be fair, also past studies meant to target real world events (even if with scales and questionnaires).

* I applaud the stress on population-based cohorts and the risk with selection bias in convenience samples. Here, it could also be mentioned that studies like the Swedish Betula project (e.g., Nyberg, et al, 2020 Ageing Res Rev) have used such an approach. In HBS, some comment is warranted on the exclusion criteria in relation to representativeness (such as taking antidepressants or having a disease that hinders physical exercise). Relatedly, the strategies to recruit participants could influence the final representativeness of the sample. For example, the involvement of employers can bias the sample to people who have a work. Have the authors any plans of assessing how representative for the general population their final sample actually will be?

* Opting out from a study and later dropping out from a study are important variables. So far, after a quite short period of time, as many as 13% withdrew (typically after the 1st assessment). I lacked information on whether the researchers will replace those who drop out at such an early stage, and also how they plan to deal with drop-out analyses.

7. PLOS authors have the option to publish the peer review history of their article (what does this mean?). If published, this will include your full peer review and any attached files.

Reviewer #1: No

---

## [Author Response · Author response to Decision Letter 0]

3 Sep 2021

Journal requirements

We have applied the PLOS ONE style templates.

2) If there are ethical or legal restrictions on sharing a de-identified data set, please explain them in detail (e.g., data contain potentially sensitive information, data are owned by a third-party organization, etc.) and who has imposed them (e.g., an ethics committee). Please also provide contact information for a data access committee, ethics committee, or other institutional body to which data requests may be sent.

Sharing the full Healthy Brain Study (HBS) resource does not comply with the EU General Data Protection Regulation, because the HBS resource contains a substantial amount of sensitive personal data. 

Applications for data access can be sent to hbs-data@radboudumc.nl. Then, the HBS scientific board reviews the application. Our governance policy ensures that all use of data is within the limits of the original purpose as described in the information brochure for participants and that only those data are shared that are relevant for answering the proposed research question. 

3) Please amend your authorship list in your manuscript file to include author Lucy Overbeek.

We would like to propose that the Healthy Brain Study consortium is the only author and omit the names of the project leader Guillén Fernández and project manager Lucy Overbeek. In our view, mentioning the consortium as the only author would perfectly reflect the nature of the team-science based cross-faculty initiative from the Radboud campus in Nijmegen. We consider our roles as supportive and required for managing the project rather than as principal investigators. We understand that a corresponding author is required. Guillén Fernández fulfills the role of the corresponding author on behalf of the consortium. As stated, all members of the consortium contributed to the design of the study, edited the manuscript, and approved the final version. 

Response to reviewer

* I agree that a reductionist approach is common, but several longitudinal studies surely include a broad range of variables and a holistic approach; this is not unique for HBS.

We fully agree with the reviewer that several longitudinal studies include a broad range of variables and a holistic approach. In lines 35-38 we mention that “we think that current brain research suffers from at least five key limitations and we set up the Healthy Brain Study (HBS) to tackle these five limitations together”. Thus, we consider the uniqueness of the HBS that it combines the following five strengths 1) a broad range of variables, 2) repeated assessments across one year, 3) single subject inferences, 4) real-world assessments, and 5) a population-based sample. 

We propose stressing that tackling these five limitations together is the uniqueness of the HBS by adapting the sentence starting in line 94 into “In conclusion, the unique feature of the HBS is that it combines the five above-mentioned strength resulting in in-depth phenotyping of a large range of cognitive, affective, behavioral, and social dimensions with a biological sampling of brain and body-related processes.”. 

However, there are other studies (e.g., Betula project) that have another focus on a longitudinal design with a very long-term perspective of more than 25 years (https://www.umu.se/en/research/projects/betula---aging-memory-and-dementia/).

* Life events are highlighted. Such events have been explored in several other past studies, and a question is whether the authors in the quite short period of 1 year expect a sufficient number of significant events to substantiate analyses? Here, the timing with the pandemic is very interesting and it is a strength of the study to capture this major event.

When designing the study, we hoped for some major national events around for example governmental elections or sports. A pandemic was of course not expected, but is indeed very interesting and offers additional opportunities.

We agree with the reviewer that one year is a rather a short period for capturing life events, but we chose the age range of 30-39 because it is a socially challenging age range that is characterized by a relatively high number of rather impactful life events (e.g., family planning, career-related changes, buying a house etc.). To assess life events, participants give responses to twenty types of life events (Brugha & Cragg, 1990). In addition, to assess the subjective impact of life events, we ask participants to rate the subjective impact the event had on their lives on a scale from 0 (no impact at all) to 10 (major impact, fundamental change). Preliminary data showed that 130 out of 298 (44%) of the HBS participants that completed the life event questionnaire at T1 experienced one or more life events in the year before participating and 78 out of 175 (45%) and 46 out of 100 (46%) of HBS participants that completed the life event questionnaire at T2 respectively T3 experienced one or more life events. As this is a protocol paper, we do not want to report any preliminary data and propose not to include the above-mentioned data on life events in the manuscript.

Reference: 

Brugha, T. S. & Cragg, D. The list of threatening experiences: the reliability and validity of a brief life events questionnaire. Acta Psychiatr Scand 82, 77-81 (1990).

* The protocol has an elegant assessment of factors such as physical activity, stress, and sleep that go beyond past studies. However, to be fair, also past studies meant to target real world events (even if with scales and questionnaires).

We agree with the reviewer that many past and ongoing studies assess real-world events (even if with scales and questionnaires). We consider our approach innovative, because for two reasons. 

First, we believe that assessing real-world events with wearable devices is more objective than scales and questionnaires. When we designed the study in 2017, hardly any longitudinal study included wearable devices. As the field of wearable technology has developed rapidly, in the meantime, several longitudinal studies have added wearable devices to their data collection methods. For example, subsets of UK Biobank participants and Rotterdam Study participants wore an accelerometer (Doherty et al., 2017; Koolhaas et al., 2017). Furthermore, in line with our response to the first comment of the reviewer, we would like to stress that including real-world assessments is one of the five strengths of the HBS, so it is not unique by itself. In particular, the HBS includes repeated assessments with wearable devices in three different seasons over one year starting at varying time points within a year. When we designed the study, to our knowledge, this was unique for HBS. In the meantime, a subset of UK Biobank participants is performing seasonal repeats with a wearable device (https://biobank.ctsu.ox.ac.uk/crystal/label.cgi?id=1008).

Second, the HBS combines physiological recordings with wearable devices with ecological momentary assessments using a smartphone application. We consider the additional collection of momentary assessments of mood and behavior and context information innovative.

We added this explanation to the discussion section when reflecting on the HBS study design (line 393).

References:

Doherty A. et al. Large Scale Population Assessment of Physical Activity Using Wrist Worn Accelerometers: The UK Biobank Study. PLoS One 12(2), e0169649 (2017).

Koolhaas C.M. et al. Objective Measures of Activity in the Elderly: Distribution and Associations With Demographic and Health Factors. Journal of the American Medical Directors Association 18 (10), 838-847 (2017).

* I applaud the stress on population-based cohorts and the risk with selection bias in convenience samples. Here, it could also be mentioned that studies like the Swedish Betula project (e.g., Nyberg, et al, 2020 Ageing Res Rev) have used such an approach. In HBS, some comment is warranted on the exclusion criteria in relation to representativeness (such as taking antidepressants or having a disease that hinders physical exercise). Relatedly, the strategies to recruit participants could influence the final representativeness of the sample. For example, the involvement of employers can bias the sample to people who have a work. Have the authors any plans of assessing how representative for the general population their final sample actually will be?

We defined a population-based sample as 1,000 HBS participants (500 men and 500 women) from the Nijmegen region (≤ 15 km) of whom 220 have a low, 340 a middle, and 430 a high level of education. We perform continuous monitoring of sex and level of education for enrollment and drop-out rates. For example, when we observed that our first wave of participants consisted of mainly higher educated people, we adapted the campaign by displaying pictures and stories of real, middle educated participants in promotion materials and focused on employers with low and middle educated staff. Concerning recruiting unemployed people, we reach them by their municipalities, our social media campaign and local influencers.

* Opting out from a study and later dropping out from a study are important variables. So far, after a quite short period of time, as many as 13% withdrew (typically after the 1st assessment). I lacked information on whether the researchers will replace those who drop out at such an early stage, and also how they plan to deal with drop-out analyses.

We aim to acquire full longitudinal datasets of 1,000 participants. We expect a withdrawal rate of 15%, and will therefore recruit 1,150 individuals to participate in the study. This information was lacking in the manuscript and has now been added to the recruitment paragraph in lines 138-139: “We aim to acquire full longitudinal datasets of 1,000 participants. We expect a withdrawal rate of 15%, and will therefore recruit 1,150 individuals to participate in the study.”.

How to deal with drop-out analyses is up to the researchers who will use the data. Partial datasets of withdrawn participants will be included in the data release, along with documentation of what parts of the data are available for each subject and information about the reasoning for withdrawal.

---

## [Decision Letter · Decision Letter 1]

15 Sep 2021

PONE-D-21-15454R1Protocol of the Healthy Brain Study: an accessible resource for understanding the human brain and how it dynamically and individually operates in its bio-social contextPLOS ONE

Dear Dr. Fernández,

Thank you for submitting your revision.

I searched for the changes but couldn't find them. Please highlight them in the edited manuscript and mention page number (and even paragraph number) in your response letter. Please make sure to edit the manuscript based on the feedback and comments you have received. 

We look forward to receiving your revised manuscript.

Kind regards,

Amir-Homayoun Javadi, PhD

Academic Editor

PLOS ONE

Journal Requirements:

Reviewers' comments:

Reviewer's Responses to Questions

**Comments to the Author**

1. Does the manuscript provide a valid rationale for the proposed study, with clearly identified and justified research questions?

Reviewer #1: Yes

2. Is the protocol technically sound and planned in a manner that will lead to a meaningful outcome and allow testing the stated hypotheses?

Reviewer #1: Yes

3. Is the methodology feasible and described in sufficient detail to allow the work to be replicable?

Reviewer #1: Yes

4. Have the authors described where all data underlying the findings will be made available when the study is complete?

Reviewer #1: Yes

5. Is the manuscript presented in an intelligible fashion and written in standard English?

Reviewer #1: Yes

6. Review Comments to the Author

You may also provide optional suggestions and comments to authors that they might find helpful in planning their study.

Reviewer #1: I thank the authors for their responses to my initial comments. I find many of their arguments valid. However, it was unclear to me what was actually changed in the manuscript. The authors mention one addition in the Discussion (not highlighted in my version of the revised ms), but other than that it was unclear from their response letter how they actually modified their manuscript. In particular, the issue of the uniqueness of their study relative to other existing ones and the difficult topic of drop-out should warrant not only sensible comments in a cover letter but also substantial changes in the ms. The same holds for life-events, even if I understand the point about not providing actual data in this particular report.

7. PLOS authors have the option to publish the peer review history of their article (what does this mean?). If published, this will include your full peer review and any attached files.

Reviewer #1: No

---

## [Author Response · Author response to Decision Letter 1]

15 Oct 2021

Dear Dr. Amir-Homayoun Javadi,

Thank you for judging our manuscript entitled ‘Protocol of the Healthy Brain Study: an accessible resource for understanding the human brain and how it dynamically and individually operates in its bio-social context’ favorably. Herewith we would like to respond to the reviewer. 

When we refer to line numbers in the following paragraphs, we refer to the line numbers in the revised manuscript with track changes.

Journal requirements

We have applied the PLOS ONE style templates.

2) If there are ethical or legal restrictions on sharing a de-identified data set, please explain them in detail (e.g., data contain potentially sensitive information, data are owned by a third-party organization, etc.) and who has imposed them (e.g., an ethics committee). Please also provide contact information for a data access committee, ethics committee, or other institutional body to which data requests may be sent.

Sharing the full Healthy Brain Study (HBS) resource does not comply with the EU General Data Protection Regulation, because the HBS resource contains a substantial amount of sensitive personal data. 

Applications for data access can be sent to hbs-data@radboudumc.nl. Then, the HBS scientific board reviews the application. Our governance policy ensures that all use of data is within the limits of the original purpose as described in the information brochure for participants and that only those data are shared that are relevant for answering the proposed research question. 

3) Please amend your authorship list in your manuscript file to include author Lucy Overbeek.

We would like to propose that the Healthy Brain Study consortium is the only author and omit the names of the project leader Guillén Fernández and project manager Lucy Overbeek. In our view, mentioning the consortium as the only author would perfectly reflect the nature of the team-science based cross-faculty initiative from the Radboud campus in Nijmegen. We consider our roles as supportive and required for managing the project rather than as principal investigators. We understand that a corresponding author is required. Guillén Fernández fulfills the role of the corresponding author on behalf of the consortium. As stated, all members of the consortium contributed to the design of the study, edited the manuscript, and approved the final version. 

Response to reviewer

* I agree that a reductionist approach is common, but several longitudinal studies surely include a broad range of variables and a holistic approach; this is not unique for HBS.

We fully agree with the reviewer that several longitudinal studies include a broad range of variables and a holistic approach. In lines 35-38 we mention that “we think that current brain research suffers from at least five key limitations and we set up the Healthy Brain Study (HBS) to tackle these five limitations together”. Thus, we consider the uniqueness of the HBS that it combines the following five strengths 1) a broad range of variables, 2) repeated assessments across one year, 3) single subject inferences, 4) real-world assessments, and 5) a population-based sample. 

We propose stressing that tackling these five limitations together is the uniqueness of the HBS by adapting the sentence starting in line 94 into “In conclusion, the unique feature of the HBS is that it combines the five above-mentioned strength resulting in in-depth phenotyping of a large range of cognitive, affective, behavioral, and social dimensions with a biological sampling of brain and body-related processes.”. 

However, there are other studies (e.g., Betula project) that have another focus on a longitudinal design with a very long-term perspective of more than 25 years (https://www.umu.se/en/research/projects/betula---aging-memory-and-dementia/).

* Life events are highlighted. Such events have been explored in several other past studies, and a question is whether the authors in the quite short period of 1 year expect a sufficient number of significant events to substantiate analyses? Here, the timing with the pandemic is very interesting and it is a strength of the study to capture this major event.

When designing the study, we hoped for some major national events around for example governmental elections or sports. A pandemic was of course not expected, but is indeed very interesting and offers additional opportunities.

We agree with the reviewer that one year is a rather a short period for capturing life events, but we chose the age range of 30-39 because it is a socially challenging age range that is characterized by a relatively high number of rather impactful life events (e.g., family planning, career-related changes, buying a house etc.). To assess life events, participants give responses to twenty types of life events (Brugha & Cragg, 1990). In addition, to assess the subjective impact of life events, we ask participants to rate the subjective impact the event had on their lives on a scale from 0 (no impact at all) to 10 (major impact, fundamental change). Preliminary data showed that 130 out of 298 (44%) of the HBS participants that completed the life event questionnaire at T1 experienced one or more life events in the year before participating and 78 out of 175 (45%) and 46 out of 100 (46%) of HBS participants that completed the life event questionnaire at T2 respectively T3 experienced one or more life events. As this is a protocol paper, we do not want to report any preliminary data and propose not to include the above-mentioned data on life events in the manuscript.

Reference: 

Brugha, T. S. & Cragg, D. The list of threatening experiences: the reliability and validity of a brief life events questionnaire. Acta Psychiatr Scand 82, 77-81 (1990).

* The protocol has an elegant assessment of factors such as physical activity, stress, and sleep that go beyond past studies. However, to be fair, also past studies meant to target real world events (even if with scales and questionnaires).

We agree with the reviewer that many past and ongoing studies assess real-world events (even if with scales and questionnaires). We consider our approach innovative, because for two reasons. 

First, we believe that assessing real-world events with wearable devices is more objective than scales and questionnaires. When we designed the study in 2017, hardly any longitudinal study included wearable devices. As the field of wearable technology has developed rapidly, in the meantime, several longitudinal studies have added wearable devices to their data collection methods. For example, subsets of UK Biobank participants and Rotterdam Study participants wore an accelerometer (Doherty et al., 2017; Koolhaas et al., 2017). Furthermore, in line with our response to the first comment of the reviewer, we would like to stress that including real-world assessments is one of the five strengths of the HBS, so it is not unique by itself. In particular, the HBS includes repeated assessments with wearable devices in three different seasons over one year starting at varying time points within a year. When we designed the study, to our knowledge, this was unique for HBS. In the meantime, a subset of UK Biobank participants is performing seasonal repeats with a wearable device (https://biobank.ctsu.ox.ac.uk/crystal/label.cgi?id=1008).

Second, the HBS combines physiological recordings with wearable devices with ecological momentary assessments using a smartphone application. We consider the additional collection of momentary assessments of mood and behavior and context information innovative.

We added this explanation to the discussion section when reflecting on the HBS study design (line 393).

References:

Doherty A. et al. Large Scale Population Assessment of Physical Activity Using Wrist Worn Accelerometers: The UK Biobank Study. PLoS One 12(2), e0169649 (2017).

Koolhaas C.M. et al. Objective Measures of Activity in the Elderly: Distribution and Associations With Demographic and Health Factors. Journal of the American Medical Directors Association 18 (10), 838-847 (2017).

* I applaud the stress on population-based cohorts and the risk with selection bias in convenience samples. Here, it could also be mentioned that studies like the Swedish Betula project (e.g., Nyberg, et al, 2020 Ageing Res Rev) have used such an approach. In HBS, some comment is warranted on the exclusion criteria in relation to representativeness (such as taking antidepressants or having a disease that hinders physical exercise). Relatedly, the strategies to recruit participants could influence the final representativeness of the sample. For example, the involvement of employers can bias the sample to people who have a work. Have the authors any plans of assessing how representative for the general population their final sample actually will be?

We defined a population-based sample as 1,000 HBS participants (500 men and 500 women) from the Nijmegen region (≤ 15 km) of whom 220 have a low, 340 a middle, and 430 a high level of education. We perform continuous monitoring of sex and level of education for enrollment and drop-out rates. For example, when we observed that our first wave of participants consisted of mainly higher educated people, we adapted the campaign by displaying pictures and stories of real, middle educated participants in promotion materials and focused on employers with low and middle educated staff. Concerning recruiting unemployed people, we reach them by their municipalities, our social media campaign and local influencers.

* Opting out from a study and later dropping out from a study are important variables. So far, after a quite short period of time, as many as 13% withdrew (typically after the 1st assessment). I lacked information on whether the researchers will replace those who drop out at such an early stage, and also how they plan to deal with drop-out analyses.

We aim to acquire full longitudinal datasets of 1,000 participants. We expect a withdrawal rate of 15%, and will therefore recruit 1,150 individuals to participate in the study. This information was lacking in the manuscript and has now been added to the recruitment paragraph in lines 138-139: “We aim to acquire full longitudinal datasets of 1,000 participants. We expect a withdrawal rate of 15%, and will therefore recruit 1,150 individuals to participate in the study.”.

How to deal with drop-out analyses is up to the researchers who will use the data. Partial datasets of withdrawn participants will be included in the data release, along with documentation of what parts of the data are available for each subject and information about the reasoning for withdrawal.

Other changes

- Line 242, Table 1a: fertility treatment and breastfeeding are not part of the questionnaire.

- Line 295: “At the end of December 2020, the HBS included 298 participants. Forty participants (13%) withdrew from the study so far, due to personal circumstances (e.g., pregnancy) or too much burden. Most participants withdrew after the first assessment. To date, participants performed 376 lab visits: 257 participants carried out the first assessment, 83 participants the second assessment, and 36 participants the third assessment completing the entire study protocol.” has been adapted into “At the end of June 2021, the HBS included 418 participants. Seventeen-one participants (17%) withdrew from the study so far, mostly because they experienced too much burden (n=41; 58%), got pregnant (n=11; 15%), or had been given a diagnosis or medication treatment (n=8; 11%). Most participants withdrew after the first assessment. To date, participants performed 755 lab visits: 380 participants carried out the first assessment, 237 participants the second assessment, and 138 participants the third assessment completing the entire study protocol.”

- Table 1a is renamed to Table 1, Table 1b to Table 7, Table 7 to Table 8.

- Reference list: references 27, 164, 165, 166 have been added.

- Link for data access is adapted to https://www.healthybrainstudy.nl/en/data-and-methods/access in line 27 and 431.

We hope we have satisfactorily responded to the comments and are looking forward to your decision about publishing our manuscript in PLOS ONE.

On behalf of the Healthy Brain Study consortium,

Guillén Fernández Lucy Overbeek

---

## [Decision Letter · Decision Letter 2]

22 Nov 2021

Protocol of the Healthy Brain Study: an accessible resource for understanding the human brain and how it dynamically and individually operates in its bio-social context

PONE-D-21-15454R2

Dear Dr. Fernández,

We’re pleased to inform you that your manuscript has been judged scientifically suitable for publication and will be formally accepted for publication once it meets all outstanding technical requirements.

Kind regards,

Fulvio D'Acquisto, PhD

Academic Editor

PLOS ONE

Reviewers' comments:

Reviewer's Responses to Questions

**Comments to the Author**

1. Does the manuscript provide a valid rationale for the proposed study, with clearly identified and justified research questions?

Reviewer #1: Yes

Reviewer #2: Yes

2. Is the protocol technically sound and planned in a manner that will lead to a meaningful outcome and allow testing the stated hypotheses?

Reviewer #1: Yes

Reviewer #2: Yes

3. Is the methodology feasible and described in sufficient detail to allow the work to be replicable?

Reviewer #1: Yes

Reviewer #2: Yes

4. Have the authors described where all data underlying the findings will be made available when the study is complete?

Reviewer #1: Yes

Reviewer #2: Yes

5. Is the manuscript presented in an intelligible fashion and written in standard English?

Reviewer #1: Yes

Reviewer #2: Yes

6. Review Comments to the Author

You may also provide optional suggestions and comments to authors that they might find helpful in planning their study.

Reviewer #1: The authors have responded well to my initial comments. I am fine with the ms in its current format and have no additional feedback to offer.

Reviewer #2: This manuscript describes the ongoing protocols employed by the HBS in Nijmegen. The study is a large-scale representative data gathering study that seeks to provide a major community resource in the form of a database of brain activation, brain structure, and cognitive function measurements. Overall, the study has been extremely well designed and is well under way. The major weakness of the study has been, in my opinion, that people in the neuroscience community are not generally aware of it. The manuscript would therefore be of potentially high impact because it would bring awareness of the study to a much wider audience. The manuscript is clear and well-written, which should add to its impact.

Comments

I realize that the manuscript has been in revision, although this is the first time I have seen it. While I believe that the manuscript is appropriate for publication in its current form, I have two comments.

First, I note that line 94 has a minor pluralization error. The word “strength,” should I believe, be “strengths.”

The second is somewhat more serious and it may be late in the study process to address this point. One of the great challenges with databases of this kind is the problem of multiple comparisons. It hundreds of scholars each perform hypothesis tests on the database, it is unclear how to deal with the problem of multiple comparisons effectively. This may be particularly acute in research conducted by citizen scientists. If the opportunity arose, I think it would be a great addition to the manuscript to include a section of statistical practices and multiple comparisons. I’m sure that the study has procedures in place for this, but more discussion would be welcome and might help other studies as well.

7. PLOS authors have the option to publish the peer review history of their article (what does this mean?). If published, this will include your full peer review and any attached files.

Reviewer #1: No

Reviewer #2: No

---

## [Editor Report · Acceptance letter]

13 Dec 2021

PONE-D-21-15454R2 

Protocol of the Healthy Brain Study: an accessible resource for understanding the human brain and how it dynamically and individually operates in its bio-social context 

Dear Dr. Fernández:

I'm pleased to inform you that your manuscript has been deemed suitable for publication in PLOS ONE. Congratulations! Your manuscript is now with our production department. 

Kind regards, 

on behalf of

Professor Fulvio D'Acquisto 

Academic Editor

PLOS ONE